# Chemical Diversity of Bastard Balm (*Melittis melisophyllum* L.) as Affected by Plant Development

**DOI:** 10.3390/molecules25102421

**Published:** 2020-05-22

**Authors:** Izabela Szymborska-Sandhu, Jarosław L. Przybył, Olga Kosakowska, Katarzyna Bączek, Zenon Węglarz

**Affiliations:** Department of Vegetable and Medicinal Plants, Institute of Horticultural Sciences, Warsaw University of Life Sciences–SGGW, 166 Nowoursynowska Street, 02-787 Warsaw, Poland; izabela_szymborska@sggw.edu.pl (I.S.-S.); jaroslaw_przybyl@sggw.edu.pl (J.L.P.); olga_kosakowska@sggw.edu.pl (O.K.); zenon_weglarz@sggw.edu.pl (Z.W.)

**Keywords:** phenolic compounds, essential oil, HPLC-DAD, GC-MS, developmental phase, age of plants, plant organs

## Abstract

The phytochemical diversity of *Melittis melissophyllum* was investigated in terms of seasonal changes and age of plants including plant organs diversity. The content of phenolics, namely: coumarin; 3,4-dihydroxycoumarin; *o*-coumaric acid 2-*O*-glucoside; verbascoside; apiin; luteolin-7-*O*-glucoside; and *o*-coumaric; *p*-coumaric; chlorogenic; caffeic; ferulic; cichoric acids, was determined using HPLC-DAD. Among these, luteolin-7-*O*-glucoside, verbascoside, chlorogenic acid, and coumarin were the dominants. The highest content of flavonoids and phenolic acids was observed in 2-year-old plants, while coumarin in 4-year-old plants (272.06 mg 100 g^–1^ DW). When considering seasonal changes, the highest content of luteolin-7-*O*-glucoside was observed at the full flowering, whereas verbascoside and chlorogenic acid were observed at the seed-setting stage. Among plant organs, the content of coumarin and phenolic acids was the highest in leaves, whereas verbascoside and luteolin-7-*O*-glucoside were observed in flowers. The composition of essential oil was determined using GC-MS/GC-FID. In the essential oil from leaves, the dominant was 1-octen-3-ol, whilst from flowers, the dominant was α-pinene.

## 1. Introduction

Bastard balm, *Melittis melissophyllum* L. (Lamiaceae), is a wild-growing perennial plant occurring in the mixed forests and bright woodlands of the western, central, and southern parts of Europe [1]. The plant has been long used in folk medicine. Its herb has been applied in the treatment of cough and sore throat, anxiety, insomnia, inflammation in the eye, as well as in wound and skin infections [2,3]. Fresh or dried leaves of the plant are used to prepare aromatic teas, which is consumed after meals to prevent digestive problems [4]. In the past, the leaves of *M. melissophyllum* were eaten as a source of vitamins and microelements [5]. Due to specific sweet odor of the dried herb of this plant, related to the presence of coumarin, in Central Europe it is used to aromatize alcohol and tobacco products. In turn, according to Maggi et al. [6], chemotypes of *M. melissophyllum* with a high content of 1-octen-3-ol may be used as a source of a mushroom-like flavoring agent for the food industry.

Previous studies on the secondary metabolites of *M. melissophyllum* have focused on the essential oils obtained from the wild-growing populations originating from Italy, Slovakia, and Spain [1,6,7,8,9]. The literature concerning phenolic compounds in *M. melissophyllum* is much scarce. So far, coumarin and some phenolic acids have been identified in the herb, namely *o*-coumaric, protocatechuic, chlorogenic, vanillic, caffeic, syringic, ferulic, sinapic, and cinnamic acids [10,11,12]. Among the flavonoids present in this plant, apigenin, kaempferol, luteolin, quercetin, and myricetin have been reported [13].

The content of biologically active compounds changes in plants depending on their age, developmental stage, and plant organ. This is also determined by the species and environmental factors influencing the growth of the plant [14,15,16]. Therefore, it is important to consider these factors when regards the quality of plants’ raw materials used in the food and cosmetics industries. Some aspects of this phenomenon, concerning seasonal changes of some phenolic compounds in *M. melisophyllus* herb have been previously reported [11,13]. However, these data are fragmented and are related only to the wild-growing plants. Strict ex situ works aiming to determine the chemical diversity of the raw material depending on the age of the plants and their developmental phase have not been conducted so far.

Therefore, in this study, our primary objective was to investigate the chemical composition of *M. melissophyllum* in terms of both seasonal changes and plant age, considering the diversity of plant organs, using high-performance liquid chromatography-diode array (HPLC-DAD) and gas chromatography-mass spectrometry (GC-MS and GC-FID).

## 2. Results and Discussion

### 2.1. Content of Phenolic Compounds in Herb Collected at Subsequent Developmental Stages of Plants

Among the analyzed compounds three flavonoids, i.e., verbascoside, apiin, and luteoline-7-*O*-glucoside were determined, with the latter as the dominant compound in this group. The content of luteolin-7-*O*-glucoside was the highest at the full flowering stage (1107.65 mg 100 g^–1^ dry weight (DW)), while the highest concentration of verbascoside and apiin was observed at the beginning of the seed-setting stage (599.86 and 156.22 mg 100 g^–1^ DW, respectively). The lowest level of all three flavonoids was detected at the beginning of the flowering stage (Table 1). Such a high content of above-mentioned flavonoids at full flowering and at the seed-setting stage might be related to the role of these compounds during a particular stage of development of the plants. It was reported that flavonoids are responsible for the transport of auxins in plants at both intra- and intercellular levels, and thus are responsible for the formation of new organs and the development of the whole plant. In addition, it was also shown that most perennial plants require flavonoids in order to set fertile seeds [17]. The high content of flavonoids observed at the generative stage of *M. melissophyllum*, during flowering and seed formation, may be associated with this phenomenon. Moreover, flavonoids are one of the most effective UV-B absorbents, serving an important role in the protection of DNA and photosynthetic systems in plants [18]. This is especially important in shade-loving plants, including *M. melissophyllum*, whose development is closely related to the access of sunlight and the quality of light. Some flavonoids, e.g., luteolin glycosides, are also capable of effectively chelating Fe or Cu-ions and thus prevent the formation of reactive oxygen species or to reduce their level [18,19]. High content of flavonoids during the flowering stage has been previously reported for many species. For this reason, the flowering stage is considered as the most appropriate one for the harvest of the herb of most of the medicinal and aromatic plants in agricultural practice.

When regards chemical changes during the ontogenic development of *M. melissophyllum*, coumarins, and phenolic acids were investigated, too. Among these, special attention was paid to the content of coumarin which is responsible for the specific, sweet aroma of the herb. The content of this compound was similar at the beginning of flowering and at the full flowering stage, and then distinctly decreased at the seed-setting stage. A similar tendency was also observed for *o*-coumaric and *p*-coumaric acids; however, their content was significantly lower. The concentration of *o*-coumaric acid 2-*O*-glucoside decreased from the beginning of flowering until the seed-setting stage. Furthermore, the content of 3,4-dihydroxycoumarin, as well as caffeic acid and cichoric acid, was the highest at full flowering stage, whereas the level of chlorogenic acid, the dominant among phenolic acids, was the highest at the beginning of the seed-setting stage (Table 1). Some aspects related to the accumulation of coumarin and phenolic acids in the leaves of *M. melissophyllum* have been investigated earlier. According to Maggi et al. [10], the content of coumarin in this raw material was the highest at the early stage of plant development, during intensive vegetative growth, and then decreased. Similar results were observed in this study, where the content of coumarin gradually decreased during observed growth of the plants (Table 1). Higher concentration of this compound at the beginning of vegetation and during flowering, when intensively growing plants are more susceptible to pathogens, is combined with its antipathogenic role [10]. It is noteworthy that the presence of coumarin in plants may also be associated with the level of shade it receives. According to Bertolucci et al. [20] the content of this compound in *Mikania laevigata,* an undergrowth species, which similarly to *M. melissophyllum* grows in forests, is affected not only by the developmental stage of the plant but also by the shade level. This was well-documented in our previous investigations on *M. melissopyllum*. The plants grown in moderate (30%) shade were found to thrive best; however, the content of coumarin in the herb was the highest in deep (50%) shade, compared to full sunlight and 30% shade [21].

Changes in the content of some phenolic acids in *M. melissophyllum* have been studied earlier in situ by Skrzypczak-Pietraszek and Pietraszek [11]. According to their results, protocatechuic acid, chlorogenic acid, *p*-hydroxybenzoic acid, vanillic acid, caffeic acid, syringic acid, *p*-coumaric acid, ferulic acid, sinapic acid, *o*-coumaric acid, and cinnamic acid were observed in higher concentration at the end of plant vegetation than at flowering stage. This, probably, was related not only to seasonal changes, but also to the age of the wild-growing plants and the environmental condition in which they grew. In our study, carried out on plants at the same age, the differences in the accumulation of phenolic acids during plant vegetation were more diverse. The dominant phenolic acid, i.e., chlorogenic acid, was observed at the highest concentration at the beginning of the seed-setting stage, whereas the content of cichoric acid was the highest at the full flowering stage (Table 1). A similar tendency for chlorogenic acid was observed previously in *Stachys officinalis* L. [22]. In some species, chlorogenic acid accumulated in distinct amounts in the seeds, is utilized during the process of germination for the deposition of phenolic polymers, e.g., lignin in cotyledonary cell walls [23]. In this study, the concentration of other phenolic acid was distinctly lower, however, it was clearly related to the developmental stage of the plant (Table 1). Previously, it has been reported, that some phenolic acids may affect the phosphorous (P) uptake of plants, and thus, influence the processes associated with flowering [24]. In general, these compounds are considered to be biotic and/or abiotic stress-dependent molecules that are important for plant’s adaptation and resistance to unfavorable environmental factors, floral induction, reproduction and senescence [17,25,26].

### 2.2. Effects of Age of Plants on the Accumulation of Phenolic Compounds

To the best of our knowledge, there are no studies regarding the relationship between the age of *M. melissophyllum* and the quality of its herb. In our experiment plant material (herb) was collected during four successive years of plant vegetation, at full flowering stage. According to our results, there were significant differences between plants of different ages with respect to the accumulation of the analyzed compounds. The flavonoids were detected in their highest quantities in 2-year-old plants and were distinctly lower in 3- and 4-year-old plants. The differences were especially visible for apiin and luteolin-7-*O*-glucoside. Similar results were obtained for the majority of the analyzed phenolic acids, i.e., chlorogenic acid, caffeic acid, ferulic acid, cichoric acid, as well as 3,4-dihydroxycoumarin and *o*-coumaric acid 2-*O*-glucoside (Table 2). This phenomenon may be related to the biology of development of this perennial plant. During the first year of vegetation, *M. melissophyllum* produces only 1–2 shoots and practically does not enter the generative phase. However, during this time, its underground organs intensively expand and in the next year (second year of vegetation), the plant produced numerous flowering shoots. The transition from vegetative to flowering phase is a crucial developmental stage influencing the reproductive success of the species. Thus, at this particular stage some species modulates their chemical profile to produce specific molecules regulating reproduction, growth, or used in defense against biotic and abiotic factors [14,15].

In this study, unlike most of the investigated compounds, the content of coumarin increased with the age of the plants and reached its maximum in 4-year-old plants (Table 2). Similar results with respect to coumarin were obtained in typical coumarin-producing grass species, namely southern sweet-grass (*Hierochloë australis* (Schrad.) Roem et Schult) [27]. Its content in the leaves increased with the age of this plant. Coumarin is reported to play an important role in inter-species competition, inhibiting seed germination. Some coumarin compounds regulate the level of chlorophyll content and thus regulate plant senescence [28].

### 2.3. Chemical Characteristics of Plant Organs

Plant diversity may be considered at the level of plant organs obtained as raw materials. Thus, in this study, flowers, leaves, shoots, and roots of *M. melissophyllum* were subjected to chemical analyzes. According to our results, the level of luteolin-7-*O*-glucoside, the dominant flavonoid, was similar in both flowers and leaves (1480.78 and 1441.36 mg 100 g^–1^ DW, respectively). In the shoots, its content was distinctly lower (20.37 mg 100 g^–1^ DW), whereas in the roots it was absent. The content of verbascoside was the highest in flowers (576.96 mg 100 g^–1^ DW), while apiin– in the leaves (134.30 mg 100 g^–1^ DW). Moreover, apiin was not detected in both shoots and roots, while roots demonstrated a distinct quantity of verbascoside (Table 3, Figure 1). The biosynthesis of flavonoids is regulated by light. Usually, they are produced in external tissues of aboveground organs. However, according to Bauer et al. [29], flavonoids are not only accumulated in the cells in which they are synthesized but may also translocate to other plant organs. As previously mentioned, flavonoids are implicated in the control of auxin transport. Thus, their presence in *M. melissophyllum* roots may be combined with auxin-dependent physiological processes taking place in the plants.

The highest content of coumarin, 3,4-dihydroxycoumarin, *o*-coumaric acid, and *o*-coumaric acid 2-*O*-glucoside, as well as chlorogenic, and cichoric acids, was detected in the leaves. In the roots, we observed distinct quantities of chlorogenic, cichoric, and ferulic acids. In addition, the roots also contained coumarin and *o*-coumaric acids (Table 3). High concentration of phenolics in *M. melissophyllum* roots may be associated with the mechanisms of plant adaptation and defense. It is well known, that plant roots interact with a diverse community of microorganisms and neighboring plants which affects different processes in the soil, including tissue decomposition and soil respiration. Roots secrete various phenolic compounds, which serve as a nutrient or acts as a toxin to those organisms [30]. They control nitrogen availability and are responsible for shifting the composition of the microbial community [31,32,33]. Phenolic compounds released from roots can also act against soil-borne pathogens and root-feeding insects [34]. As a typical allelopathic substance phenolics reduce the germination of seeds of competing plants, inhibit their root elongation, and cell division and interfere with their normal growth and development [35]. Among compounds with the above-mentioned activity the most common are chlorogenic, benzoic, gallic, ferulic, coumaric, *p*-hydroxybenzoic, caffeic acids, and coumarin, along with some flavonoids, namely: catechin, naringenin, and quercetin [36,37,38].

Essential oils are another group of compounds important when regards the quality of herbal raw materials of *M. melissophyllum*, although their content in this plant is relatively low. According to our study, the content of essential oil in its leaves was higher than in flowers (0.09 and 0.03 g 100 g^−1^, respectively). The GC-MS analysis showed that the composition of essential oils distilled from these organs was significantly different. The dominant compound in the leaves was 1-octen-3-ol, while in the flowers it was α-pinene. In total, 22 compounds were identified in flowers and 27– in the essential oil obtained from leaves, which accounted for 95.63 and 90.98%, respectively, of the total identified substances (Table 4). Apart from coumarin, the essential oil determines the specific aroma of *M. melissophyllum* herb. It is worth noting that coumarin might be a component of the essential oils, too. The chemical composition of volatiles present in the dried leaves, stems, calyx, and corolla of *M. melissophyllum* was previously investigated by Maggi et al. [9]. According to their results, the emission of volatiles from leaves was over thrice higher than that of other organs, and the dominant compound, i.e., 1-octen-3-ol was similar in quantities in the leaves, stems, and calyx. However, it was not detected in corolla. It was observed that 1-octen-3-ol is present in plant raw materials in low concentrations; the compound is formed in higher quantities only during hydrodistillation [6]. Our study, carried out on essential oils distilled from dried raw materials, have shown that 1-octen-3-ol as a dominant compound of essential oil from leaves (29.19%), was present in essential oil from flowers, too (1.04%). Flowers predominantly contained α-pinene (66.58%). The percentage share of coumarin in the essential oil of leaves was relatively low (1.17%), whereas in the essential oil from flowers the compound was not detected (Table 4). This stays in an agreement with previously published results showing that in the essential oil obtained from flowering herb the content of coumarin is ≥1% [6]. Due to a specific aroma of *M. melissophyllum* herb, it is nowadays used to aromatize alcohol and tobacco products, and is considered as an interesting source of mushroom-like odor for the food industry.

## 3. Materials and Methods

### 3.1. Field Experiment

The field study was carried out in the experimental station at Warsaw University of Life Sciences–SGGW (WULS-SGGW) (Warsaw, Poland). The seedlings of *M. melissophyllum* were grown in a greenhouse and planted out at a spacing 50 × 40 cm, at the beginning of October 2015. The experiment was conducted from 2015 to 2019. Plant materials for the analysis were collected starting from 2016 (first year of field vegetation).

### 3.2. Plant Material

#### 3.2.1. Content of Phenolic Compounds in Herb Collected at Subsequent Developmental Stages of Plants

The raw material (herb) was collected from 3-year-old plants at three consecutive stages: at the beginning of flowering, at the full flowering stage, and at the seed-setting stage. At each stage, the herb was collected from ten randomly selected plants, dried in the dark at 35 °C, and subjected to chemical analysis via HPLC-DAD.

#### 3.2.2. Effects of Age of Plants on the Accumulation of Phenolic Compounds

The raw material (herb) was collected from 1-, 2-, 3-, and 4-year-old plants at the full flowering stage. The herb was cut 5 cm above ground level from ten randomly selected plants, then dried in the dark at 35 °C and subjected for chemical analysis via HPLC-DAD.

#### 3.2.3. Chemical Characteristics of Plant Organs

Flowers, leaves, stems, and roots were collected separately from 3-year-old plants at the full flowering stage. They were dried at 35 °C and subjected to chemical analysis via HPLC-DAD. The chemical composition of the essential oil obtained from leaves and flowers was carried out using GC-MS and GC-FID.

### 3.3. Chemical Analysis

#### 3.3.1. Chemicals

All standards were purchased from ChromaDex^®^ (California, LA, USA). Acetonitrile and methanol were of analytical and HPLC grade (Merck KGaA, Darmstadt, Germany). Water was purified by using a water purification system (Thermo Scientific, Pacific 20 AFT, Barnstead, New Hampshire, USA).

#### 3.3.2. HPLC-DAD

Extraction of the raw material was carried out using Büchi Labortechnik AG Extraction System B-811 (Flawil, Switzerland). Briefly, 1.000 g of air-dried, finely powdered raw material was extracted with 100 mL of methanol. Soxhlet hot extraction with 25 cycles, flushing, and drying was applied. After evaporation of solvent, the residue was dissolved in 10 mL of methanol. The extracts were filtered with Supelco Iso-Disc^TM^ Syringe Tip Filter Unit, PTFE membrane (Merck KGaA, Darmstadt, Germany) and subjected to HPLC analysis. The analysis was performed using a Shimadzu chromatograph, equipped with an auto-sampler SIL-20A, photodiode array detector SPD-M10A VP PDA, and CLASS VP™ 7.3 chromatography software (Shimadzu, Kyoto, Japan). Separations were achieved by using a C18 reverse-phase Kinetex^™^ column, 2.6 μm, 100 mm × 4.60 mm, with a porous outer layer on solid silica core particles (Phenomenex^®^, Torrance, CA, USA). Validation parameters has already been described in detail by Szymborska-Sandhu et al. [21]. Quantification of the analyzed compounds was performed with analytical wavelength appropriate for each compound: (1) coumarin, 3,4-dihydroxycoumarin, *o*-coumaric, acid and *o*-coumaric acid 2-*O*-glucoside were quantified at 276 nm; (2) *p*-coumaric acid was quantified at 309 nm; (3) chlorogenic acid, caffeic acid, ferulic acid, cichoric acid were quantified at 325 nm; (4) verbascoside was quantified at 330 nm; (5) apiin was quantified at 336 nm; (6) luteolin-7-*O*-glucoside was quantified at 347 nm. Standard curve parameters were calculated with Microsoft Excel.

#### 3.3.3. Isolation of Essential Oils and GC-MS and GC-FID Analysis

The essential oils were obtained in accordance with the European Pharmacopoeia 9th edition [39]. Briefly, 20 g of air-dried raw material was subjected to hydrodistillation in a Clevenger-type apparatus for 3 h. Qualitative GC-MS and quantitative GC-FID analysis of the essential oils obtained from flowers and leaves were performed by usage an Agilent Technologies 7890A gas chromatograph equipped with a flame ionization detector (FID) and MS Agilent Technologies 5975C Inert XL_MSD with Triple Axis Detector (Agilent Technologies, Wilmington, DE, USA). Capillary, polar column HP 20M (25 m × 0.32 mm × 0.3 µm film thickness) (Agilent Technologies, Wilmington, DE, USA) was applied. Detailed procedure, including separation conditions and identification of compounds was conducted as described earlier by Bączek et al. [40].

### 3.4. Statistical Analysis

The results obtained were analyzed by using Statgraphics Plus version 4.1 (Statgraphics Technologies Inc., The Plains, Virginia) with a one-way analysis of variance (ANOVA). Tukey’s test at a significance level of α = 0.05 was applied. All the chemical analyses were conducted in triplicates, and the data were expressed as mean ± standard deviation (SD). Differences between individual means were deemed to be significant at *p* < 0.05.

## 4. Conclusions

According to the requirements of the phytopharmaceutical and food industries, the quality of herbal raw materials is strictly associated with the high and stable content of secondary metabolites influencing the biological activity of these materials. Such quality standards are very hard to be met, especially in the case of wild-growing plants that are collected from the genetically unstable populations, from plants of different ages, that grow in diverse environments. In this study, some factors influencing the accumulation of phenolic compounds in the raw materials of *M. melissophyllum*, were investigated. The results showed that both the age and the developmental stage of the plants strongly influence the content of phenolic compounds in *M. melissophyllum* herb. The differences in the content and composition of biologically active compounds among plant organs were also significant. Thus, these factors should be taken into consideration in plant production. Our results may be useful not only in the identification of quality markers of these raw materials but also in the works concerning the introduction of the plant into cultivation, including the selection of uniform lines that may be used in industries.

## Figures and Tables

**Figure 1 molecules-25-02421-f001:**
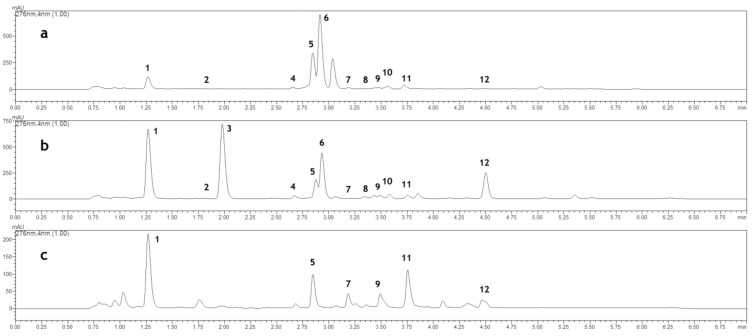
HPLC chromatogram of: (**a**) flowers extract (**b**) leaves extract (**c**) roots extract. The following compounds were analyzed: (**1**) chlorogenic acid; (**2**) caffeic acid; (**3**) *o*-coumaric acid 2-*O*-glucoside; (**4**) *p*-coumaric acid; (**5**) verbascoside; (**6**) luteolin 7-*O*-glucoside; (**7**) ferulic acid; (**8**) 3,4-dihydroxycoumarin; (**9**) cichoric acid; (**10**) apiin; (**11**) *o*-coumaric acid; (**12**) coumarin.

**Table 1 molecules-25-02421-t001:** Content of phenolic compounds in herb collected at subsequent developmental stages of plants (mg 100 g^–1^ DW).

Compounds	Developmental Stages					
	Beginning of Flowering		Full Flowering		Seed-Setting	
Flavonoids:						
Verbascoside	265.50	±14.66 _c_	338.82	±26.45 _b_	599.86	±27.00 _a_
Apiin	46.77	±6.67 _c_	97.92	±12.37 _b_	156.22	±6.03 _a_
Luteolin-7-*O*-glucoside	555.61	±35.35 _c_	1107.65	±47.64 _a_	773.32	±32.30 _b_
Coumarins and Phenolic Acids:						
Coumarin	201.12	±13.37 _a_	220.27	±10.35 _a_	145.58	±12.70 _b_
3,4-Dihydroxycoumarin	40.28	±3.38 _a_	76.06	±5.61 _b_	45.11	±3.55 _a_
*o*-Coumaric acid	33.48	±6.64 _a_	29.76	±8.15 _a_	9.88	±1.27 _b_
*o*-Coumaric acid 2-*O*-glucoside	370.09	±43.52 _a_	297.51	±56.00 _a_	122.09	±30.93 _b_
*p*-Coumaric acid	3.69	±0.77 _a_	2.87	±0.44 _a_	0.79	±0.21 _b_
Chlorogenic acid	151.19	±22.15 _c_	412.46	±19.85 _b_	545.00	±23.00 _a_
Caffeic acid	4.47	±0.23 _c_	6.82	±0.65 _a_	5.50	±0.79 _b_
Ferulic acid	7.87	±1.30 _a_	7.26	±0.63 _a_	9.92	±1.28 _a_
Cichoric acid	63.57	±7.54 _b_	89.81	±15.29 _a_	52.46	±12.22 _c_

Values marked in rows with different letters differ at *p* < 0.05.

**Table 2 molecules-25-02421-t002:** The influence of age of plants on accumulation of investigated phenolic compound (mg 100 g^–1^ DW).

Compounds	Age of Plants							
	1-Year-Old		2-Year-Old		3-Year-Old		4-Year-Old	
Flavonoids:								
Verbascoside	442.24	±35.31 _b_	576.68	±29.64 _a_	438.65	±17.42 _b_	472.30	±31.03 _b_
Apiin	164.13	±9.50 _b_	243.19	±5.46 _a_	88.55	±6.10 _d_	117.52	±6.13 _c_
Luteolin-7-*O*-glucoside	1774.79	±22.35 _b_	2601.95	±36.07 _a_	1456.12	±39.11 _d_	1557.98	±32.03 _c_
Coumarins and phenolic acids:								
Coumarin	164.62	±10.34 _c_	204.99	±8.36 _b_	219.16	±11.20 _b_	272.06	±12.44 _a_
3,4-Dihydroxycoumarin	70.00	±2.98 _c_	175.07	±2.75 _a_	99.94	±3.71 _b_	62.65	±3.05 _c_
*o*-Coumaric acid	66.02	±4.45 _a_	33.04	±3.71 _b_	34.70	±3.60 _b_	30.60	±4.35 _b_
*o*-Coumaric acid 2-*O*-glucoside	326.32	±44.89 _d_	521.03	±37.61 _a_	459.44	±30.50 _b_	400.10	±31.90 _c_
*p*-Coumaric acid	3.39	±0.42 _b_	3.27	±0.31 _b_	2.95	±0.69 _b_	5.71	±0.26 _a_
Chlorogenic acid	446.46	±20.00 _b_	686.58	±23.00 _a_	419.93	±25.36 _b_	223.43	±22.92 _c_
Caffeic acid	6.49	±0.50 _b_	12.22	±0.34 _a_	5.28	±0.52 _b_	6.11	±0.48 _b_
Ferulic acid	4.50	±0.69 _b_	11.11	±1.24 _a_	5.64	±1.48 _b_	3.28	±1.24 _b_
Cichoric acid	90.91	±8.32 _b_	136.65	±10.15 _a_	89.24	±7.11 _b_	85.90	±11.42 _b_

Values marked in rows with different letters differ at *p* < 0.05.

**Table 3 molecules-25-02421-t003:** Diversity among plants organs concerning the content of investigated phenolic compound (mg 100 g^–1^ DW).

Compounds	Plant Organs							
	Flowers		Leaves		Shoots		Roots	
Flavonoids:								
Verbascoside	576.96	±27.78 _a_	396.55	±34.41 _b_	88.84	±7.40 _d_	279.53	±8.56 _c_
Apiin	45.86	±0.68 _b_	134.30	±9.60 _a_	n.d.		n.d.	
Luteolin-7-*O*-glucoside	1480.78	±106.15 _a_	1441.36	±66.38 _a_	20.37	±3.05 _b_	n.d.	
Coumarins and phenolic acids:								
Coumarin	6.39	±0.29 _d_	164.04	±3.44 _a_	63.85	±5.15 _b_	24.83	±0.33 _c_
3,4-Dihydroxycoumarin	19.94	±0.30 _b_	64.48	±0.96 _a_	5.60	±0.23 _c_	n.d.	
*o*-Coumaric acid	1.33	±0.08 _d_	25.88	±0.18 _a_	5.25	±0.36 _c_	6.49	±0.10 _b_
*o*-Coumaric acid 2-*O*-glucoside	n.d.		420.20	±6.60 _a_	34.50	±0.06 _c_	n.d.	
*p*-Coumaric acid	5.37	±0.34 _a_	3.01	±0.26 _b_	1.52	±0.32 _c_	n.d.	
Chlorogenic acid	105.84	±4.29 _d_	358.41	±25.73 _a_	152.41	±10.88 _c_	241.84	±13.73 _b_
Caffeic acid	14.02	±1.38 _a_	7.16	±0.09 _b_	4.28	±0.21 _c_	n.d.	
Ferulic acid	15.86	±0.65 _b_	7.68	±0.39 _c_	3.57	±0.42 _d_	50.89	±4.28 _a_
Cichoric acid	27.53	±0.74 _b_	92.97	±8.65 _a_	14.64	±0.32 _c_	89.91	±3.36 _a_

Values marked in rows with different letters differ at *p* < 0.05; n.d.–not detected.

**Table 4 molecules-25-02421-t004:** Gas chromatographic composition (% peak area) of essential oils.

No.	Chemical Compound	RI *^a^*	Leaves	Flowers
1	α-Pinene	1027	1.38	66.58
2	Camphene	1074	0.00	0.67
3	Hexanal	1082	0.27	0.00
4	β-Pinene	1115	0.10	6.80
5	2-Hexenal, (E)-	1139	1.56	0.00
6	Heptanal	1186	0.00	1.08
7	Cosmene	1204	0.00	0.27
8	Furan, 2-pentyl-	1234	0.39	0.00
9	m-Cymene	1275	0.05	0.37
10	2,3-Octanedione	1320	0.37	0.00
11	*cis*-Rose oxide	1353	0.00	2.03
12	Nonanal	1392	0.13	0.00
13	1-Octen-3-ol	1443	29.19	1.04
14	α-Campholenal	1496	0.00	1.74
15	Linalol	1541	0.99	0.00
16	β-caryophyllene	1593	0.87	0.00
17	Terpinen-4-ol	1599	0.00	0.24
18	β-terpinyl acetate	1624	0.47	1.22
19	Myrtenal	1632	0.00	0.73
20	Verbenol	1659	0.00	0.66
21	α-Terpineol	1698	0.00	0.67
22	Heptadecane	1701	0.00	0.74
23	2-Methyl coumarone	1725	1.54	0.00
24	Citronellol	1763	0.00	0.74
25	Myrtenol	1790	0.00	1.10
26	Caryophyllene oxide	1795	3.78	0.32
27	6,8-Nonadien-2-one	1823	2.37	0.00
28	*trans*-Carveol	1837	0.00	0.66
29	Geraniol	1839	1.37	0.65
30	Dehydrodihydroionone	1896	0.49	0.00
31	β-Ionone	1937	3.65	0.00
32	Globulol	2033	0.72	0.00
33	Ledol	2047	7.31	0.00
34	(-)-Spathulenol	2126	2.79	0.00
35	Phyton	2164	0.00	1.74
36	Aromadendrene oxide-(2)	2232	1.42	0.00
37	Farnesyl acetate	2259	1.53	0.00
38	Coumarin	2475	1.17	0.00
39	Phytol	2614	8.78	0.00
40	Tetradecanoic acid	2686	1.95	0.00
41	Heptacosane	2700	0.00	0.93
42	n-Hexadecanoic acid	2912	20.99	0.00
Total identified			95.63	90.98
Aliphatics			56.83	3.79
Terpenoids			31.56	87.19
Monoterpene hydrocarbons			1.53	74.69
Oxygenated monoterpens			2.83	10.44
Sesquiterpenes hydrocarbons			0.87	0.00
Oxygenated sesquiterpenes			17.55	0.32
Diterpenes			8.78	1.74
Aromatics			2.71	0.00
Others			4.53	0.00
Content of essential oil (g 100 g^−1^ DW)			0.09	0.03

*^a^* RI–retention index calculated on polar column.

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
