# Peer review of "Chemical Diversity of Bastard Balm (Melittis melisophyllum L.) as Affected by Plant Development"

_molecules, 2020, doi:10.3390/molecules25102421_

Round 1

Reviewer 1 Report

The manuscript entittled "Chemical diversity of bastard balm (Melittis mel-2 isophyllum L.) as affected by plant development" is well presented and justified.

Three minor suggestions are given to improve the readability of the manuscript:

  1. First time that the importance of the shade in phenols are mention, it would be desirable to mention how. It is inferred later in the manuscript, but a clarification should help the readability of the document.
  2. Extraction, purification... methods, even when referred to a reference, could be brefly described for a better understanding
  3. In general, coumarin and related compounds are highlited along the manuscript and described more specifically than other compounds; it is suggested to include in the background a sentence explaining the higher importance of these components.

In general, the manuscript is understandable and clear.

Author Response

Thank You very much for the review. All the suggestions have been included in the revised text of the manuscript.

  1. Results and Discussion chapter has been supplemented with additional information on the impact of shade on the accumulation of coumarin in M  melissophyllum. The following information has been introduced:

Lines 104-107: “This was well-documented in our previous investigations on M. melissopyllum. The plants grown in moderate (30%) shade were found to thrive best; however, the content of coumarin in the herb was the highest in deep (50%) shade, compared to full sunlight and 30% shade [21].”

  1. We have modified the description of the HPLC-DAD analysis. The following description has been introduced:

“Extraction of the raw material was carried out using Büchi Labortechnik AG Extraction System B-811 (Flawil, Switzerland). Briefly, 1.000 g of air-dried, finely powdered raw material was extracted with 100 mL of methanol. Soxhlet hot extraction with 25 cycles, flushing and drying was applied. After evaporation of solvent, the residue was dissolved in 10 mL of methanol. The extracts were filtered with Supelco Iso-DiscTM Syringe Tip Filter Unit, PTFE membrane (Merck KGaA, Darmstadt, Germany) and subjected to HPLC analysis. The analysis was performed using a Shimadzu chromatograph, equipped with an auto-sampler SIL-20A, photodiode array detector SPD-M10A VP PDA, and CLASS VP™ 7.3 chromatography software (Shimadzu, Kyoto, Japan). Separations were achieved by using a C18 reversed-phase Kinetex column, 2.6 μm, 100 mm × 4.60 mm, with a porous outer layer on solid silica core particles (Phenomenex®, Torrance, CA, USA). Validation parameters has already been described in details by Szymborska-Sandhu et al. [21]. Quantification of the analyzed compounds was performed with analytical wavelength appropriate for each compound: (1) coumarin, 3,4-dihydroxycoumarin, o-coumaric acid and o-coumaric acid 2-O-glucoside were quantified at 276 nm; (2) p-coumaric acid was quantified at 309 nm; (3) chlorogenic acid, caffeic acid, ferulic acid, cichoric acid were quantified at 325 nm; (4) verbascoside was quantified at 330 nm; (5) apiin was quantified at 336 nm; (6) luteolin-7-O-glucoside was quantified at 347 nm. Standard curve parameters were calculated with Microsoft Excel.”

  1. High importance of coumarin in M. melissophyllum herb was stressed both in Introduction and Results and Discussion chapters. The following information has been introduced:

Introduction – Lines 36-38: “Due to specific sweet odor of the dried herb of this plant, related to the presence of coumarin, in Central Europe it is used to aromatize alcohol and tobacco products.”

Results and Discussion – Lines 84-87: “When regards chemical changes during the ontogenic development of M. melissophyllum, coumarins and phenolic acids were investigated, too. Among these, special attention was paid to the content of coumarin which is responsible for the specific, sweet aroma of the herb.”

Reviewer 2 Report

The manuscript "Chemical diversity of bastard balm (Melittis melisophyllum L.) as affected by plant development" deal with the changes in phenolic composition and chemical composition of the essential oil in function of phenoligical stages and plant development. The plant subject of this study is used in the folk medicine of several countries. The study is well designed and well conducted, and the manuscript well written. Essential oils and methanolic extract of the different plant organs were used for chemical analysis, collecting samples in different phenological stages for 1 year, and for 4 years to evaluate age influence on chemical composition. Analytical methods for identification and quantification are well described and the conclusions are coherent with results. 

I suggest to authors only very minor amendments:

- Please describe the plant material extraction procedure instead of to refer only to an already published article. In this way it is easier for readers understand rapidily which kind of extraction was used. For details you can refer to the exhisting literature.

- Please check Table 4 for a more correct use of the italic style and for some typos (i.e. "Pinen")

- Please add the yields of the essential oils. Although the authors report that the yield is low, it would be better if it were reported at least in the tables.

Author Response

Thank You very much for the review. All the suggestions have been included in the revised text of the manuscript.

  1. We have modified the description of the HPLC-DAD analysis. The following description has been introduced:

Extraction of the raw material was carried out using Büchi Labortechnik AG Extraction System B-811 (Flawil, Switzerland). Briefly, 1.000 g of air-dried, finely powdered raw material was extracted with 100 mL of methanol. Soxhlet hot extraction with 25 cycles, flushing and drying was applied. After evaporation of solvent, the residue was dissolved in 10 mL of methanol. The extracts were filtered with Supelco Iso-DiscTM Syringe Tip Filter Unit, PTFE membrane (Merck KGaA, Darmstadt, Germany) and subjected to HPLC analysis. The analysis was performed using a Shimadzu chromatograph, equipped with an auto-sampler SIL-20A, photodiode array detector SPD-M10A VP PDA, and CLASS VP™ 7.3 chromatography software (Shimadzu, Kyoto, Japan). Separations were achieved by using a C18 reversed-phase Kinetex column, 2.6 μm, 100 mm × 4.60 mm, with a porous outer layer on solid silica core particles (Phenomenex®, Torrance, CA, USA). Validation parameters has already been described in details by Szymborska-Sandhu et al. [21]. Quantification of the analyzed compounds was performed with analytical wavelength appropriate for each compound: (1) coumarin, 3,4-dihydroxycoumarin, o-coumaric acid and o-coumaric acid 2-O-glucoside were quantified at 276 nm; (2) p-coumaric acid was quantified at 309 nm; (3) chlorogenic acid, caffeic acid, ferulic acid, cichoric acid were quantified at 325 nm; (4) verbascoside was quantified at 330 nm; (5) apiin was quantified at 336 nm; (6) luteolin-7-O-glucoside was quantified at 347 nm. Standard curve parameters were calculated with Microsoft Excel.

  1. We have checked the use of italics and typos in Table 4 and provided changes which are marked with red font in the text.

  1. The yields of the essential oils of leaves and flowers have been added. It was provided at the end of table 4 (marked with red font). The following text was added in lines 197-198: “According to our study, the content of essential oil in its leaves was higher than in flowers (0.09 and 0.03 g 100 g -1, respectively).”